# The Impact of Therapeutic Community Gardening on the Wellbeing, Loneliness, and Life Satisfaction of Individuals with Mental Illness

**DOI:** 10.3390/ijerph192013166

**Published:** 2022-10-13

**Authors:** Carly J. Wood, Jo L. Barton, Claire L. Wicks

**Affiliations:** 1School of Sport, Rehabilitation and Exercise Sciences, University of Essex, Colchester CO4 3SQ, UK; 2School of Health and Social Care, University of Essex, Colchester CO4 3SQ, UK

**Keywords:** green social prescribing, therapeutic community gardening, mental ill-health, wellbeing, health

## Abstract

Background: literature on the mental health benefits of therapeutic community gardening is not specific to individuals with mental illness and reports short-term outcomes. The impact of the coronavirus pandemic on intervention effectiveness is also unknown. This study examined the impact of therapeutic community gardening prior to and across the pandemic on the wellbeing of individuals referred for support with their mental illness. Methods: garden members (*n* = 53; male = 36, female = 17) aged 47.38 ± 13.09 years reported their wellbeing at baseline and four follow-up points (FU1–FU4) across the pandemic. Results: there was significant quadratic growth in wellbeing (−1.248; *p* < 0.001) that varied between genders (*p* = 0.021). At baseline, male wellbeing scores were significantly lower (*p* = 0.020) than the UK population norm, but there were no significant differences at any other follow-up point. Female wellbeing was significantly lower than the UK population norm at baseline (*p* < 0.001), FU1 (*p* = 0.012) and FU2 (*p* < 0.001), but not FU3 and FU4. Conclusion: therapeutic community gardening can improve and maintain the wellbeing of individuals with mental illness, even when wellbeing is deteriorating nationally. Future research should further demonstrate the long-term and cost-effectiveness of interventions.

## 1. Introduction

Mental health is defined as “a state of well-being in which every individual realises their own potential, can cope with the normal stresses of life, can work productively and fruitfully, and is able to make a contribution to her or his community” [1]. A mental illness is characterized by a clinically significant disturbance in an individual’s cognition, emotional regulation, or behaviour, that is associated with distress or impairment in important areas of functioning, such as work, daily activities, or personal relationships [2]. Globally in 2019, 1 in 8 people were living with a mental illness [2]. In the UK one in six individuals experience a common mental illness such as anxiety or depression at any time [3] with mental health problems being one of the main causes of disease burden worldwide [4] and costing the UK economy an estimated £117.9 billion annually [5].

Social connections are vital for human wellbeing across the life span and influence mental health [6]. The absence of fulfilling social relationships and social contact can result in loneliness, which is defined as ‘the unpleasant experience that occurs when a person’s network of social relations is deficient in some important way’ [7]. Weiss [8] distinguishes between two separate components of loneliness; emotional loneliness which relates to the absence of an intimate relationship, for example with a close friend or partner, and social loneliness, which relates to the absence of positive engagement with wider social networks. Loneliness is associated with increased morbidity and mortality [9], hypertension, increased cortisol levels in both the morning and evening, and poorer sleep quality [10]. Loneliness is also associated with an increased risk of mental illness [10,11], whilst individuals with a common mental illness are also more likely to experience loneliness [12]. In fact, evidence suggests that each one-point increase in loneliness is associated with a 16% increase in severity of depression, with 11–18% of cases of depression potentially being prevented if loneliness were eliminated [13]. Furthermore, increased experiences of loneliness are associated with significantly lower life satisfaction [14].

In 2018, the UK Government launched its first loneliness strategy [15], which highlighted that one approach to tackling loneliness may be through community-based interventions. This is closely aligned with the social prescribing agenda, which offers a means by which third sector organisations can provide non-medical sources of support such as nature-, sport-, or art-based therapies, to address mental, psychosocial, and socioeconomic needs [16]. The National Health Service (NHS) has committed to diversifying the range of social prescriptions to ensure accessibility to services across the UK, aiming for over 900,000 people to be referred to schemes by 2023/2024 [17]. Green social prescribing is one type of social prescription which aims to improve health and wellbeing through exposure to, and interaction with natural environments [18]. The importance of access to green space for health and the inequality of access was highlighted by the COVID-19 pandemic. In response to this, in July 2020 the UK Government invested £5.77 million in green social prescribing to prevent and tackle the increasing prevalence of mental illness and poor mental health [19].

Therapeutic community gardening is a specific type of green social prescription which is focused on using garden space and gardening activities to help people improve their mental health, build social skills, and develop confidence; with qualified therapist input or mental health support [20]. However, within the literature there is considerable heterogeneity between interventions with many studies focusing on community or allotment gardening without mental health support [20]. Furthermore, gardening interventions are often complex, comprising multiple components such as access to nature, social interaction, and physical activity, which makes it difficult for researchers to untangle which components might be most beneficial, or to identify whether health benefits occur through a combination or interaction of multiple components [20].

In their systematic review of allotment gardening, Genter et al. [21] identified five distinct pathways through which gardening could improve health and wellbeing; (i.) via offering a stress-relieving refuge (e.g., offering a calm, relaxing environment, and social contact as a buffer against stress); (ii.) by supporting a healthier lifestyle (e.g., increased physical activity and healthier dietary choices); (iii.) through provision of social opportunities (e.g., establishing friendships and social interaction, offering respite from personal agony); (iv.) by providing valued contact with nature (e.g., observing change and nurturing plants); and (v.) enabling self-development (e.g., increased self-confidence, skills, and reframing attitudes). This is supported by the literature demonstrating significant improvements in depression, anxiety, quality of life, mindfulness, life satisfaction, social cohesion, and happiness in individuals with mental ill-health following therapeutic community gardening interventions [22,23]; and the reported benefits of gardening interventions for the general population which include reductions in depression, anxiety, psychological distress, and loneliness; and enhanced quality of life, happiness and community belonging [20,21,24,25,26,27,28]. It also aligns with research reporting that stakeholders perceive that the mental health benefits of therapeutic community gardening are derived from the opportunities for interaction and connection with nature, the development of social support and relationships, fostering a sense of belonging; and the hope it provides for the future [29,30,31].

Despite a range of literature indicating the health benefits of partaking in allotment, community and therapeutic gardening, there is a lack of data specifically focused on individuals suffering from mental illness and the use of therapeutic gardening as a green social prescription. In addition, much of the evidence base reports the health benefits following engagement in one gardening session, or short-term interventions [20]. For therapeutic community gardening and green social prescribing to be upscaled and more widely available for the treatment of mental illness, specific evidence of the mental health benefits for individuals suffering with mental illness is required. Evidence of the longer-term protective effects would also further enhance the evidence base.

The environmental context of the coronavirus pandemic must also be considered. Throughout the pandemic, wellbeing declined in the general population [32,33], along with increased rates of loneliness, social isolation, and mental illness [34,35,36,37]. In the first year of the pandemic, depression and anxiety were reported to have increased by more than 25% [38]. There were also significant disruptions to the availability of mental health services in 93% of countries worldwide, which may have adversely impacted the mental health of service users and the effectiveness of interventions [39,40]. Data collected over the period of the pandemic should therefore be interpreted and considered in terms of this wider environmental context. The primary aim of this study was to examine the impact of therapeutic community gardening prior to and across the coronavirus pandemic on the wellbeing of individuals referred for the treatment of their mental illness. The secondary aim was to track changes in loneliness and life satisfaction across the pandemic in individuals referred to therapeutic community gardening for the treatment of their mental illness.

## 2. Materials and Methods

### 2.1. Participants and Programme Details

Participants (*n* = 53) were adults (aged 18 years+) attending a therapeutic community gardening programme to support their mental ill-health. The therapeutic community gardening programme (called Growing Together (GT)) was delivered by Trust Links, an independent mental health and wellbeing charity who provide therapeutic gardens across Essex, UK. Garden beneficiaries are referred to as members to create an inclusive environment in which everyone is equally valued. Participants will therefore be referred to as members throughout this manuscript. Members were individuals who had been attending any of the GT garden sites for any duration and had a range of mental health diagnoses (undisclosed to the researchers). Members were typically referred to GT via GPs, community psychiatric nurses, social prescribing link workers, job centres and self-referral. When attending, GT members took part in various gardening activities including sowing seeds, potting, and general garden maintenance. When on site, garden members were supported by horticultural project workers who lead on gardening activities, mental health staff who support the mental health of members; and volunteers, who are members of the public recruited to support members, engage with the project, and maintain the gardens.

During the coronavirus pandemic, the gardens were closed from March to June 2020, when they reopened for groups of six for 90-min sessions. In April 2021 this increased to two-hour sessions for 10 members and in July 2021 three-hour sessions took place with 10 members. From October 2021 there were no restrictions in place; members could attend for full days without limitations on numbers, as they did prior to the pandemic. GT members received mental health support from Trust Links staff via telephone and zoom when the gardens were closed throughout the pandemic and whilst the service was reopening.

### 2.2. Procedure

Members of the GT therapeutic community gardens were asked by members of GT mental health staff whether they would be willing to complete a short survey about their attendance at GT and the impact on wellbeing, loneliness, and life satisfaction at three timepoints; between May 2021 and January 2022. Members were recruited on a rolling basis, with new members being invited to participate as they joined GT and existing members invited as they returned to the gardens following the easing of coronavirus restrictions.

For existing members, the timepoint 1 (T1) survey was conducted between May and October 2021, the timepoint 2 (T2) survey between July and November 2021; and the timepoint 3 (T3) survey between December 2021 and January 2022. Completion of the surveys was via hardcopy and was voluntary. All members were provided with a participant information sheet detailing the study procedures and provided informed consent prior to taking part. GT mental health staff were trained on consent and survey procedures by the researchers, and assisted members with these processes where an individual required support with reading or writing. Survey data was linked with each members unique Trust Links ID number and entered onto the Trust Links password protected Charitylog database where member information is stored and managed. Anonymised data containing participants ID, and survey information was subsequently downloaded for analysis; no identifiable information was provided to the researchers. Only Trust Links could match members to their ID number. Ethical approval for the study was granted by the School of Sport, Rehabilitation and Exercise Sciences Ethics Sub-committee at the University of Essex (Ref: ETH2021-0911).

In addition to the data collected at the three survey timepoints, Trust Links provided secondary wellbeing data (see Section 2.3.2) for members that had previously completed surveys as part of Trust Links routine service evaluations. This was collected between July 2016 and August 2020. All participants provided consent for the data to be shared with a third party for research purposes. Member’s Trust Links ID was used to link primary and secondary data.

### 2.3. Measures

#### 2.3.1. Demographic and Attendance Information

On the first primary data collection survey members were asked to provide their date of birth and gender and to identify how long they had been attending GT, their typical frequency of attendance and whether they had attended in the last year during the coronavirus pandemic. At all primary data collection survey timepoints members were also asked to identify what they perceived were the most through to least important aspects of GT from: (i.) the interaction with nature; (ii.) participation in physical activity; (iii.) social interaction and (iv.) skill and knowledge development.

#### 2.3.2. Wellbeing (Secondary Data Provided)

Members’ wellbeing in the last month was assessed via the short form Warwick Edinburgh Mental Wellbeing Scale (SWEMWBS) [41]. The SWEMWBS consists of seven positively worded items from the full 14-item scale and is scored by summing responses to each item, which are scored on a five-point Likert scale from 1 (none of the time) to 5 (all of the time). The raw SWEMWBS scores were converted to metric scores [42] prior to data analyses; with scores ranging from 7 to 35 and higher scores indicating better wellbeing. The SWEMWBS has been reported to have a Cronbach’s alpha of 0.84 using England population-level data [43]; with correlations between the full and short version being 0.954 [42]. In the current sample, the average Cronbach alpha across the wellbeing timepoints was 0.899, indicating very good internal consistency. The UK normative figures in the general population are 23.67 and 23.59 for males and females, respectively [43].

#### 2.3.3. Loneliness

Members perceived experience of loneliness (social and emotional) was assessed via the six-item version of the De Jong Gierveld loneliness scale [44]. Emotional loneliness is calculated by summing items 1, 4 and 6, with the five response options scored either 1 or 0 (1 = ‘Yes!’, ‘Yes’ or ‘More or less’; 0 = ‘No’ or ‘No!’) and social loneliness calculated by summing items 2, 3 and 5 with the five response options scored either 1 or 0 (0 = ‘Yes!’ or ‘Yes’; 1 = ‘More or less’, ‘No’ or No!). Scores for emotional and social loneliness range from 0 to 3; with higher scores representing greater experiences of loneliness. The measure is reported to be reliable in adult populations with Cronbach’s alphas from 0.67 to 0.74 for the emotional loneliness sub-scale, and 0.70 to 0.73 for the social loneliness sub-scale [44]. In the current sample the average Cronbach alpha across the timepoints was 0.579 and 0.791 for emotional and social loneliness, respectively. A score of between one and three on both subscales is considered to represent loneliness, whilst a score below one indicates that the member is ‘not lonely’ [45].

#### 2.3.4. Life Satisfaction

Life satisfaction was measured using a single item measure that asked members to rate how satisfied they felt with their life on a scale from 0 to 10, with 0 = ‘not at all satisfied’ and 10 = ‘completely satisfied’. This question was adopted by the Office for National Statistics in 2013 and has been widely benchmarked in national panel surveys [46,47] and national wellbeing and happiness comparisons [48,49]. It has also been used to explore the impact of a range of nature-based interventions on mental health and wellbeing [50]. The average life satisfaction score in the general population between April 2020 and March 2021 was 7.39 [51].

### 2.4. Data Processing and Statistical Analysis

#### 2.4.1. Wellbeing

Primary and secondary wellbeing data were combined for analysis. Data was treated as a ‘baseline’ measure of wellbeing if it reflected participants wellbeing when they first joined GT. Subsequent measures of wellbeing including both primary and secondary data were combined in date order to reflect wellbeing across multiple consecutive timepoints. Overall, wellbeing data is included for 53 members, 43 of which had a baseline measure of wellbeing with four additional follow up wellbeing measurements being included in the analysis. These are referred to as follow up one (FU1, *n* = 53), follow up two (FU2, *n* = 50), follow up three (FU3, *n* = 37) and follow up four (FU4, *n* = 22) to distinguish from the remaining study data (i.e., loneliness and life satisfaction) which was only collected at three primary data collection timepoints. The average duration between wellbeing measures at baseline and FU1 was 0.86 years, FU1 and FU2 1.70 years, FU2 and FU3 0.44 years and FU3 and FU4 0.27 years. Data therefore reflects tracking of wellbeing over an average of 3.27 years.

An MCAR test was used to examine whether the missing wellbeing data were missing completely at random. Scores across the measurement time points, along with age were entered as quantitative variables and gender was added as a categorical variable. A non-significant chi square indicated that data was missing completely at random.

Latent Growth Modelling was used to examine changes in wellbeing over time by gender. Within-person change across the timepoints were initially examined by specifying latent intercept and linear slopes for each of the dependent variables and estimating the means and variances. Quadratic growth factors were subsequently added to explore the extent to which the rate of change itself changes over time. One samples t-tests were also conducted to compare the wellbeing scores for males and females at each timepoint to the UK population norms of 23.67 for males and 23.59 for females [43].

#### 2.4.2. Loneliness and Life Satisfaction

Data for loneliness and life satisfaction were measured at the three primary data collection timepoints (between May 2021 and January 2022). This data only reflects baseline data for four members who joined GT at the first primary data collection survey timepoint and cannot be considered a reflection of baseline values for the entire sample. The timepoints for measures of loneliness and life satisfaction are therefore referred to as timepoint 1 (T1), timepoint 2 (T2) and timepoint 3 (T3) to distinguish from the combined primary and secondary wellbeing data. The average duration between T1 and T2 was 0.24 years and between T2 and T3 was 0.29 years. An MCAR test was used to examine whether the missing data for each variable was missing completely at random. Score across the measurement time points, along with age were entered as quantitative variables and gender was added as a categorical variable. A non-significant chi indicated that data was missing completely at random.

For data tracking purposes, means and standards deviations (SD) for loneliness and life satisfaction scores in males and females at each of the three timepoints are presented, along with the overall mean. Multiple imputation was conducted to account for missing data; the pooled imputed mean is also presented alongside the raw values. For consistency, wellbeing data for these three timepoints is also presented. One samples t-tests were conducted to compare life satisfaction scores at each timepoint to the UK normative score of 7.39 for 2020/2021 [51].

## 3. Results

### 3.1. Participants

The overall sample included 53 adults aged 47.38 ± 13.09 years; with 67.9% (*n* = 36) of the sample being males. At the first primary data collection timepoint (May–October 2021), the majority of members had been attending GT for between 1 and 3 years and regularly attended once per week (Table 1). At the second primary data collection timepoint, 51 members completed surveys with 68.6% (*n* = 35) being males. At the final primary data collection timepoint 38 members completed surveys, with 63.2% (*n* = 24) being males. At all timepoints the majority of members (T1 = 57.8%, T2 = 52.9%, T3 = 52.6%) perceived social interaction as the most important element of GT; whilst at timepoint 1 (35.6%) and timepoint 2 (35.3%) skill and knowledge development were the least important element of GT. At timepoint 3 participation in physical activity was deemed the least important factor (31.6%).

### 3.2. Wellbeing

An MCAR test with the wellbeing scores at the five measurement timepoints and age added as quantitative variables, and gender added as a categorical variable revealed that the data were missing completely at random (*X*^2^ (21) = 28.36; *p* = 0.130). This enabled us to conduct latent growth modelling without concerns over missing data.

Latent Growth Modelling revealed that the model represented a good fit, with all fit indices meeting specified cut points: *X*^2^ (8) = 7.166; *p* = 0.519; CFI = 1.00, TLI = 1.01; SRMR = 0.061, RMSEA (90% CI) = 0.00 (0.00–0.15). The intercept model was statistically different from zero (21.54; *p* < 0.001); with no significant variation in the intercept model between individuals (5.039; *p* = 0.603). The linear slope alone was not statistically significant (0.024; *p* = 0.941), with no variation in the linear slope between individuals (−0.564; *p* = 0.508).

The addition of the quadratic growth curve resulted in a significant linear (3.121; *p* = 0.002) and quadratic (−1.248; *p* < 0.001) growth. There was no significant variation in members linear (0.516; *p* = 0.974) or quadratic (−0.500; *p* = 0.785) change over time. Covariances between the intercept and linear slope (4.99; *p* = 0.640), intercept and quadratic slope (−1.06; *p* = 0.804) and linear and quadratic (0.510; *p* = 0.922) slopes were also non-significant. There were no differences between males and females in the intercept (−1.773; *p* = 0.103); or linear slope (−3.67; *p* = 0.054); but there were significant gender differences in the quadratic slope (1.639; *p* = 0.021). Male improvements in wellbeing occurred earlier in the intervention but plateaued with time; whilst female wellbeing fluctuated over time, with the greatest score at follow up 3 (Figure 1).

One samples t-tests revealed that baseline male wellbeing scores were significantly lower (*t*(30) = −2.448; *p* = 0.020) than the mean population score of 23.67 reported by Fat et al. [43]; however, this was not the case at any other follow-up timepoint (FU1 *p* = 0.690, FU2 *p* = 0.486; FU3 *p* = 0.501; FU4 *p* = 0.176; see Figure 1). Female wellbeing was also significantly lower than the mean population score of 23.59 [43] at baseline (*t*(11) = −5.269; *p* < 0.001), FU1 (*t*(16) = −2.826; *p* = 0.012) and FU2 (*t*(14) = −5.542; *p* < 0.001). Scores at FU3 (*p* = 0.181) and FU4 (*p* = 0.060) were not significantly different from the UK mean population score.

Wellbeing data collected at T1–T3 only, in line with loneliness and life satisfaction, are presented in Table 2. In males, scores were not significantly different to the UK population norm at any timepoint (T1 *p* = 0.984; T2 *p* = 0.074; T3 *p* = 0.718). In females, scores at T1 (*t*(14) = −5.758; *p* < 0.001) and T2 (*t*(14) = −3.042; *p* = 0.009) were significantly lower than the UK population norm, but this was not the case at T3 (*p* = 0.262).

### 3.3. Loneliness and Life Satisfaction

An MCAR test with the loneliness and life satisfaction scores at the three timepoints and age added as a quantitative variable, and gender added as a categorical variable revealed that the data were missing completely at random: emotional loneliness (*X*^2^ (14) = 16.46; *p* = 0.286), social loneliness (*X*^2^ (14) = 11.05; *p* = 0.682), life satisfaction (*X*^2^ (5) = 2.751; *p* = 0.738). As data were missing at random, we used multiple imputation to calculate pooled means for the missing data.

Table 2 presents the raw mean and SD for loneliness and life satisfaction variables at each timepoint, along with the pooled imputed means. Scores for emotional and social loneliness were highest at T2, representing greater experiences of loneliness. Scores at T1 and T3 were similar, with females experiencing greater social loneliness than males at all timepoints, but only greater emotional loneliness than males at T1 and T2. Female emotional loneliness declined across the timepoints, whilst male emotional loneliness increased. Conversely, male social loneliness declined whilst females’ social loneliness remained stable. Life satisfaction was highest at T2 and lower in females at all timepoints. Male scores increased across the timepoints, whilst female scores increased at T2 but decreased slightly at T3. One samples t-tests revealed that life satisfaction for all participants combined was significantly lower than the mean population score of 7.39 [47] at T1 (*t*(52) = −2.955; *p* = 0.005), but not at T2 (*p* = 0.103) or T3 (*p* = 0.059).

## 4. Discussion

The primary aim of this study was to examine the impact of therapeutic community gardening prior to and across the coronavirus pandemic on the wellbeing of individuals referred for the treatment of their mental illness. Overall, members wellbeing scores improved, with significant linear and quadratic growth over the course of the evaluation period, which was on average 3.27 years. This is in line with literature demonstrating the benefits of therapeutic community gardening for individuals with mental illness [22,23] and the benefits of gardening interventions more broadly [20,21,27,28,29,30,31]. Furthermore, this study further adds to the evidence base, presenting novel evidence of trajectories of change in wellbeing as a result of long-term engagement with a green social prescribing intervention. The findings also revealed significant differences in quadratic growth between males and females. In line with the literature, male wellbeing was higher than female wellbeing at all timepoints [32,43]. Males also experienced greater improvements in wellbeing earlier in the intervention, with scores peaking at follow up 1 (an average of 0.86 years after baseline); whilst female scores fluctuated more gradually over time and peaked at follow up 3 (an average of 3.00 years after baseline).

Male wellbeing was significantly lower than the UK population normative value of 23.67 [43] at baseline but not at any other timepoints; whilst female wellbeing was significantly lower than the UK normative score of 23.59 at baseline, FU1 and FU2. Whilst these comparisons are likely to be influenced by sample size reductions over time, it may be that the therapeutic community gardening intervention was more successful at improving and maintaining wellbeing in males. Milligan et al. [52] suggest that many community activities tend to be viewed as gender-neutral, ignoring the idea that men and women may engage with, or perform activities within different environments, in different ways. It may also be that the multiple components of therapeutic community gardening and their potential interactions differ by gender, resulting in varying health impacts. In line with this, Hoglund et al. [53] reported that men derive more benefits through productive activities, such as gardening, whilst the literature on the Men’s Shed movement indicates significant benefits for male mental wellbeing due to ‘male-friendly’ approaches which enable men to be involved in development of community activities from the outset [54,55,56]. This co-production approach is at the heart of therapeutic community gardening; and thus, may explain differences between male and female responses. Therapeutic community gardening and green social prescribing organisations should therefore consider how to optimise the outcomes of interventions for both males and females.

Members changes in wellbeing over time should also be considered in the context of the coronavirus pandemic. Although wellbeing is expected to fluctuate in response to life events [57], the gradual reduction in male wellbeing in the later stages of the intervention and the fluctuating wellbeing scores in women might be a direct result of the pandemic. All data collected at follow up points two, three and four was collected between August 2020 and January 2022, during the coronavirus pandemic. If the pandemic had not occurred it is possible that improvements in wellbeing would have continued or stabilised over time, in line with the research of Smyth et al. [58] demonstrating sustained wellbeing increases after 13 months of attending nature-based volunteering programmes. However, it is important to note that neither male or female wellbeing returned to baseline values, despite the pandemic, and throughout the pandemic wellbeing scores decreased in the general population [32]. In a sample of 932 UK adults surveyed during the first coronavirus lockdown, 36.8% of whom had poor mental health, Smith et al. [32] reported average wellbeing scores as low as 21.5 ± 45.2 and 20.6 ± 4.8 for males and females, respectively, figures significantly lower than the values reported by Fat et al. [43]. Wood et al. [33] also reported a significantly lower wellbeing score of 21.5 ± 3.5 during the pandemic in a sample of 315 UK adults. Thus, if the data in the current study are considered in line with these revised values; it could be speculated that the therapeutic community gardening intervention, even at the reduced capacity with which it operated throughout the pandemic, supported the maintenance of members wellbeing at a time where scores were decreasing nationally.

The secondary aim of this study was to track changes in loneliness and life satisfaction across the pandemic in individuals referred to therapeutic community gardening for the treatment of their mental illness. Member’s loneliness and life satisfaction were tracked over three timepoints, with the first timepoint between May and October 2021 and T2 and T3 an average of 0.24 years and 0.53 years after T1. For social loneliness, scores were lowest at T3 and highest at T2, whilst emotional loneliness was lowest at T1 and highest at T2. The slight reduction in social loneliness at T3 may be a result of the significant easing of coronavirus restrictions announced for January 2022 and socialisation over the Christmas period. Furthermore, in October 2021 all restrictions at the gardens were removed, meaning that members could attend for a full day and that there were no limits on numbers of members present. This might have increased social interactions and thus reduced social loneliness. If we consider social loneliness scores using published cut points [45] which indicate a score of greater than one equates to loneliness; then members decrease in social loneliness at T3 moves them into the ‘not socially lonely’ category, whilst scores at T1 and T2 are just above the cut-off. Given that population social interaction was limited throughout the pandemic, the maintenance and reduction in social loneliness in members may positively reflect on the critical social interaction derived through garden attendance. This in line with literature demonstrating improvements in social cohesion following therapeutic community gardening [22] and by garden members rating the social interaction at the gardens as the most important element of GT at all survey timepoints. Given that loneliness increases annual GP visits by 1.8-fold and Accident and Emergency visits by 1.6-fold [59]; it is possible that the reductions in loneliness experienced through green social prescribing programmes such as GT may reduce public health costs. Using data from a range of nature-based interventions, Pretty et al. [50] predicted these savings to be £714 after one year and £5317 after 10 years. However, the explanations for changes in social isolation documented are speculative and based on short term tracking of loneliness data.

In contrast to social loneliness, emotional loneliness was lowest at T1 and highest at T2, with participants being classed as ‘emotionally lonely’ at all timepoints, according to the cut points of Van Tilburg and Gierveld [45]. The fluctuations in emotional loneliness are likely to be reflective of life outside of the GT gardens, as this type of loneliness is focused on the absence of an intimate relationship [8], which is less likely to be targeted through therapeutic gardening interventions. However, the pandemic may have prevented members interaction with close friends and partners; and thus, could explain the fluctuations in these scores.

Life satisfaction increased between T1 and T2, with a slight reduction at T3; however overall scores improved between T1 and T3; in line with literature supporting improved life satisfaction and quality of life following participation in therapeutic community gardening [22,23]. Scores were significantly lower than the UK national average for 2020/2021 at T1, but not at T2 or T3. Literature indicates that it is hard and costly to make changes in life satisfaction of +1 pt at the population level, with positive and negative changes occurring as a result of significant life events such as marriage, divorce and job loss usually being less than one point [46,47,48,49]. In fact, over a decade, only five countries increased life satisfaction by more than +1 pt [46,47]. Between T1 and T2 and T1 and T3, members scores increased by mean values of 0.58 and 0.44, respectively. The cost implications of these changes in life satisfaction could be significant. For example, it is reported that a 1-point increase in life satisfaction is more economically beneficial to individuals who have low life satisfaction. For individuals on a median income of £23,000, a 1-point increase in life satisfaction from 6 to 7, is suggested to be equal to an extra £7140 of income [60]. Given that member’s life satisfaction at T1 was in the bottom 23% of the population and significantly lower than the UK norm [51], it is feasible to assume the documented changes in life satisfaction may provide economic benefits.

Some literature evaluating social prescribing programmes has also indicated reduced public service use following engagement, for example after one-year such programmes have been reported to reduce GP appointments by 15–25% [61]. Pretty and Barton [50] estimated that this could prevent approximately £831 of public health costs per person after one year and £6456 per person after ten years. Furthermore, when combining the potential cost savings with those from reduced loneliness and increased life satisfaction, the economic returns may be between £6000 and £14,000 per person after one year, and £8600 and £24,500 per person after ten years [50]. However, these figures do not consider the costs of running the programme. Pretty et al. [50] estimate that the total benefits to costs for the GT therapeutic community gardening programme specifically to be 6.42 after one year and 7.61 after ten years. This data further supports the potential benefits of green social prescribing schemes.

The current study has a number of limitations. First, there was a dropout of participants over time and smaller sample sizes in females which may have reduced the power of the comparisons undertaken. At the first primary data collection timepoint (May-October 2021) 53 members completed a survey, reducing to 38 members at T3. This is equivalent to a dropout rate of 28.3% and is in line with dropout rates reported in psychosocial interventions (27.2%) for people with severe mental illness [62]. Despite the smaller sample sizes in females, the dropout rate for females was only 17.6% compared to 33% in males (17.6%). The data in the current study were also missing completely at random. However, the reasons for participant dropout from the study are unknown. It may be that participants simply left the programme or no longer wished to take part in the study; it would be beneficial to obtain this information and to interview participants to identify reasons for leaving the programme, as this would help to determine barriers to attendance and inform future practice, delivery, upscaling, and policy decision making. The lack of ‘true’ baseline data for loneliness and life satisfaction is also a limitation as it may have resulted in an underestimation of the magnitude of health benefits that individuals with mental ill-health derive from therapeutic community gardening. The coronavirus pandemic has also had significant health and wellbeing implications worldwide and is likely to have significantly influenced members wellbeing, life satisfaction and loneliness throughout the evaluation period. Although Trust Links provided mental health support throughout the pandemic period, it is not possible to determine to what extent the pandemic impacted outcomes. Future research should ensure that measures of mental health and wellbeing are collected immediately upon entry to social prescribing programmes and that regular tracking takes place over time; to further build on the long-term data generated in the current study. This long-term tracking, alongside evidence of the cost-effectiveness of such interventions, is essential to further demonstrating the mental health benefits of green social prescriptions and thus influencing mental health policy and practice.

## 5. Conclusions

Overall, the findings of this study revealed that attendance at GT therapeutic community gardens resulted in significant linear and quadratic growth in wellbeing over time, with differences in quadratic growth between males and females. Both male and female wellbeing improved over time and did not return to baseline levels. Loneliness and life satisfaction also fluctuated over the course of the pandemic; with life satisfaction scores only being lower than the UK normative value at T1 and members being deemed ‘not socially lonely’ at T3. These improvements in outcome measures occurred during the coronavirus pandemic when wellbeing was decreasing nationally. In order to further demonstrate the mental health benefits of therapeutic community gardening for individuals with mental illness, continued long term tracking of wellbeing variables along with collection of data immediately upon entry to programmes is required. Evidence of the cost-effectiveness of a range of green social prescriptions would also strengthen the evidence base and inform mental health policy and practice.

## Figures and Tables

**Figure 1 ijerph-19-13166-f001:**
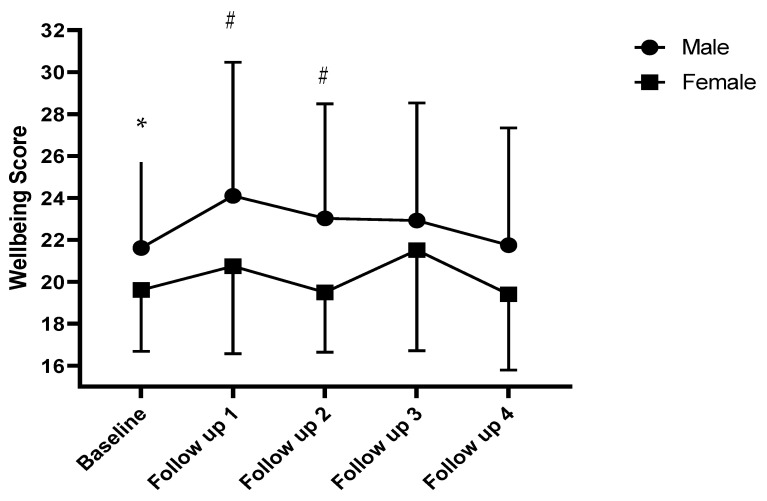
Mean ± SD of changes in wellbeing in male and female garden members (* indicates significantly lower score than UK population norm for males and females; # indicates significantly lower score than the UK population norm for females only). Note: follow up 1 is on average 0.86 years after baseline, follow up 2 1.70 years after follow up 1, follow up three 0.44 years after follow up 2 and follow up four 0.27 years after follow up 3.

**Table 1 ijerph-19-13166-t001:** Member attendance information at timepoint one.

		Percentage (*n*)
Duration of attendance	Just started	8.2% (*n* = 4)
	Up to one year	2.0% (*n* = 1)
	1–3 years	36.7% (*n* = 18)
	4–6 years	22.4% (*n* = 11)
	7–9 years	20.4% (*n* = 10)
	10+ years	10.2% (*n* = 5)
Frequency of attendance	Once a week	63.8% (*n* = 30)
	Twice a week	29.8% (*n* = 14)
	3–4 times a week+	6.4% (*n* = 3)
Attended in the last year	Yes	93.3% (*n* = 42)
	No	6.7% (*n* = 3)

Note: excluding new members four members did not specify their duration of attendance or whether they had attended in the last year at timepoint 1 and two members did not specify their frequency of attendance.

**Table 2 ijerph-19-13166-t002:** Mean ± SD raw data and pooled mean data for outcome measures.

		Timepoint 1	Timepoint 2	Timepoint 3
Wellbeing (7–35 scale)	Male	23.69 ± 5.90 (*n* = 35)	22.17 ± 4.66 (*n* = 33)	23.22 ± 6.00 (*n* = 24)
	Female	18.76 ± 3.25 * (*n* = 15)	20.72 ± 3.65 * (*n* = 15)	21.94 ± 4.83 (*n* = 12)
	Total	22.21 ± 5.69 (*n* = 50)	21.72 ± 4.39 (*n* = 48)	22.80 ± 5.60 (*n* = 36)
	Pooled (*n* = 53)	21.98	21.46	22.74
Emotional Loneliness (0–3 scale)	Male	1.40 ± 1.03 (*n* = 35)	1.77 ± 0.96 (*n* = 34)	1.70 ± 1.15 (*n* = 23)
	Female	2.09 ± 0.83 (*n* = 11)	1.80 ± 1.01 (*n* = 15)	1.36 ± 1.21 (*n* = 11)
	Total	1.57 ± 1.03 (*n* = 46)	1.78 ± 0.96 (*n* = 49)	1.59 ± 1.16 (*n* = 34)
	Pooled (*n* = 53)	1.55	1.79	1.58
Social Loneliness (0–3 scale)	Male	0.83 ± 1.16 (*n* = 36)	0.94 ± 1.08 (*n* = 35)	0.57 ± 0.95 (*n* = 23)
	Female	1.46 ± 1.39 (*n* = 13)	1.40 ± 1.30 (*n* = 15)	1.46 ± 1.39 (*n* = 13)
	Total	1.00 ± 1.24 (*n* = 49)	1.08 ± 1.16 (*n* = 50)	0.89 ± 1.19 (*n* = 36)
	Pooled (*n* = 53)	1.02	1.12	0.86
Life satisfaction (0–10 scale)	Male	6.89 ± 2.92 (*n* = 36)	7.00 ± 2.60 (*n* = 35)	7.04 ± 2.82 (*n* = 24)
	Female	4.97 ± 1.82 (*n* = 17)	6.38 ± 2.36 (*n* = 16)	5.43 ± 3.08 (*n* = 14)
	Total	6.27 ± 2.75 * (*n* = 53)	6.80 ± 2.52 (*n* = 51)	6.45 ± 2.90 (*n* = 38)
	Pooled (*n* = 53)	6.27	6.85	6.71

* Indicates a significantly lower score than the UK population norm. Note: the average duration between T1 and T2 was 0.24 years and between T2 and T3 was 0.29 years.

## Data Availability

The data presented in this study are available via the UK Data Service Repository on request from the corresponding author.

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
