# Peer review of "The Impact of Therapeutic Community Gardening on the Wellbeing, Loneliness, and Life Satisfaction of Individuals with Mental Illness"

_ijerph, 2022, doi:10.3390/ijerph192013166_

Round 1

Reviewer 1 Report

This study is valuable insofar as it provides information on the buffering effect offered by therapeutic community gardening to mental ill population when exposed to highly stressful situations, such as the COVID19 pandemic. This quasi-experimental scenario has made it possible to show the therapeutic benefits of this type of intervention.

 The study is well planned, and the writing of the article is clear and precise. However, two issues have come to my attention that I think should be clarified.

Line 128 - 130   it is stated: “When on site, garden members were supported …by mental health staff who supported the mental health of members”. It is not specified in the article if during the period of partial reintroduction to the garden activity (of one and half-, two-, and three-hours sessions) (lines 132- 134) the members had psychological support and if so, to what point this support received during the pandemic could have been mainly responsible for the wellbeing observed in the sample in comparison with the general population

In line 117-   only information on the total number of participants is given, however specific numbers should appear, since:

 In lines 437- 441:  in limitations of the study the total number of participating members, at the beginning and at the end of primary data measurements, is given. Although a dropout rate of 26.4% is in line with dropout rates reported in other psychosocial interventions, it is stated that the dropout of the members over time and smaller sample sizes in females may have reduced the power of the comparisons undertaken.   How big has been the dropout in women? Is the final number adequate to offer reliability to the comparisons? Since the study is comparing by sex please specify the number of male and female participants in each sex group at the time of the first primary data measurement and at the time of the last measurement.

Author Response

Comment

Response

This study is valuable insofar as it provides information on the buffering effect offered by therapeutic community gardening to mental ill population when exposed to highly stressful situations, such as the COVID19 pandemic. This quasi-experimental scenario has made it possible to show the therapeutic benefits of this type of intervention.
The study is well planned, and the writing of the article is clear and precise. However, two issues have come to my attention that I think should be clarified.

Thank you very much for your positive comments and suggestions to improve the manuscript, which we have addressed below.

Line 128 - 130   it is stated: “When on site, garden members were supported …by mental health staff who supported the mental health of members”. It is not specified in the article if during the period of partial reintroduction to the garden activity (of one and half-, two-, and three-hours sessions) (lines 132- 134) the members had psychological support and if so, to what point this support received during the pandemic could have been mainly responsible for the wellbeing observed in the sample in comparison with the general population

Additional information has been added in the methods section to highlight the support that Trust Links members received throughout the pandemic (lines 155-157). This has also been reflected upon in the discussion section (lines 505-507).

In line 117-   only information on the total number of participants is given, however specific numbers should appear, since: In lines 437- 441:  in limitations of the study the total number of participating members, at the beginning and at the end of primary data measurements, is given. Although a dropout rate of 26.4% is in line with dropout rates reported in other psychosocial interventions, it is stated that the dropout of the members over time and smaller sample sizes in females may have reduced the power of the comparisons undertaken.   How big has been the dropout in women? Is the final number adequate to offer reliability to the comparisons? Since the study is comparing by sex please specify the number of male and female participants in each sex group at the time of the first primary data measurement and at the time of the last measurement.

Thank you for this suggestion. We have added some detail regarding changes in participant numbers over time (by sex) into the results section (lines 287-289) and added further discussion of dropout in the relevant section of the discussion (lines 492-494).

Reviewer 2 Report

First of all, thank you for the ability to review the manuscript,

It's a good job, on a very interesting topic such as the impact of community therapeutic gardening on the well-being, loneliness and life satisfaction of people with mental illness, but before publication should incorporate the following suggestions, I beg the authors to follow them one by one, they are small things that will help improve the manuscript,

some important references are missing in the introduction,

the discussion section should be improved and more worked, in my opinion, that there is so much data, difficult to read fluid, the authors can think how to express the same but without so much numerical figure, which hinders a good discussion 

The bibliography of the previous introductory studies should be related to the discussion, which will give more power to the manuscript, try to link it

with these changes made, the article will improve and have the quality to be published in this prestigious journal,

finally, congratulate the authors for a very good work

Author Response

Comment

Response

First of all, thank you for the ability to review the manuscript. It's a good job, on a very interesting topic such as the impact of community therapeutic gardening on the well-being, loneliness, and life satisfaction of people with mental illness, but before publication should incorporate the following suggestions, I beg the authors to follow them one by one, they are small things that will help improve the manuscript,

Thank you very much for your positive comments and suggestions to improve the manuscript which we have addressed below.

Some important references are missing in the introduction,

Thank you for your comment. We have added some additional material in the introductory section (lines 89-99).

The discussion section should be improved and more worked, in my opinion, that there is so much data, difficult to read fluid, the authors can think how to express the same but without so much numerical figure, which hinders a good discussion

Thank you for this suggestion. We have attempted to streamline the discussion to improve readability, reducing numerical data where possible. In places in this data is key to the discussion of the potential economic impacts of social prescribing activities and has therefore been kept in the manuscript.

The bibliography of the previous introductory studies should be related to the discussion, which will give more power to the manuscript, try to link it

Thank you for this very helpful comment. We have revisited the discussion section and ensured that we have linked back to the material in the introduction where possible.

finally, congratulate the authors for a very good work

Thank you very much for your positive feedback.

Reviewer 3 Report

Thank you for the opportunity to review the article " The impact of therapeutic community gardening on the wellbeing, loneliness, and life satisfaction of individuals with mental illness ". Examining wellbeing, loneliness, and life satisfaction of individuals with mental illness is an important area of research, particularly bearing in mind the current relevance of COVID-19. However, the quality of ideas and methods of this paper is inadequate. There are several aspects that I ponder could add to improve the quality of the manuscript.

Abstract

Analytical methods (i.e., statistical methods) need not be specified in abstract. However, the sample size, percent women, and average age with range should be specified.

Introduction

Literature review is needed. In addition, the contributions of the paper is unclear. What are contributions to previous literatures (Theoretical implications)? 

Result

Data analysis needs to be strengthened. In figure 1, Mean ± SD can’t indicate significant between men and women. 

Author Response

Comment

Response

Thank you for the opportunity to review the article " The impact of therapeutic community gardening on the wellbeing, loneliness, and life satisfaction of individuals with mental illness ". Examining wellbeing, loneliness, and life satisfaction of individuals with mental illness is an important area of research, particularly bearing in mind the current relevance of COVID-19. However, the quality of ideas and methods of this paper is inadequate. There are several aspects that I ponder could add to improve the quality of the manuscript.

Thank you very much for your positive comments and suggestions to improve the manuscript which we have addressed below.

Analytical methods (i.e., statistical methods) need not be specified in abstract. However, the sample size, percent women, and average age with range should be specified.

Thank you for this suggestion. Analytical methods have been removed from the abstract and the suggested information included.

Introduction. Literature review is needed. In addition, the contribution of the paper is unclear. What are contributions to previous literatures (Theoretical implications)?

Thank you for this comment. We have added some additional literature in the introductory section (lines 89-99) and added some information on the theoretical and practical implications of this work in the discussion section.

Result. Data analysis needs to be strengthened. In figure 1, Mean ± SD can’t indicate significant between men and women.

We have made some edits to the results section to improve readability. We had previously indicated statistically significant differences in figure 1 but these disappeared during formatting. We have added these in this updated version.

Round 2

Reviewer 3 Report

Table 1 (line 295), add (%) after Percentage and remove % from 8.2% to 6.7%.